# Delivery of Alpha-Mangostin Using Cyclodextrins through a Biological Membrane: Molecular Dynamics Simulation

**DOI:** 10.3390/molecules25112532

**Published:** 2020-05-29

**Authors:** Wiparat Hotarat, Bodee Nutho, Peter Wolschann, Thanyada Rungrotmongkol, Supot Hannongbua

**Affiliations:** 1Center of Excellence in Computational Chemistry (CECC), Department of Chemistry, Faculty of Science, Chulalongkorn University, Bangkok 10330, Thailand; h.wiparat@gmail.com (W.H.); bnutho@gmail.com (B.N.); 2Department of Pharmaceutical Chemistry, University of Vienna, 1090 Vienna, Austria; karl.peter.wolschann@univie.ac.at; 3Institute of Theoretical Chemistry, University of Vienna, 1090 Vienna, Austria; 4Program in Bioinformatics and Computational Biology, Graduate School, Chulalongkorn University, Bangkok 10330, Thailand; 5Structural and Computational Biology Research Unit, Department of Biochemistry, Faculty of Science, Chulalongkorn University, Bangkok 10330, Thailand; 6Molecular Sensory Science Center, Faculty of Science, Chulalongkorn University, Bangkok 10330, Thailand

**Keywords:** alpha-mangostin, cellular membrane, POPC, cyclodextrins, drug penetration, molecular dynamics, PMF

## Abstract

α-Mangostin (MGS) exhibits various pharmacological activities, including antioxidant, anticancer, antibacterial, and anti-inflammatory properties. However, its low water solubility is the major obstacle for its use in pharmaceutical applications. To increase the water solubility of MGS, complex formation with beta-cyclodextrins (βCDs), particularly with the native βCD and/or its derivative 2,6-dimethyl-β-CD (DMβCD) is a promising technique. Although there have been several reports on the adsorption of βCDs on the lipid bilayer, the release of the MGS/βCDs inclusion complex through the biological membrane remains unclear. In this present study, the release the MGS from the two different βCDs (βCD and DMβCD) across the lipid bilayer was investigated. Firstly, the adsorption of the free MGS, free βCDs, and inclusion complex formation was studied by conventional molecular dynamics simulation. The MGS in complex with those two βCDs was able to spontaneously release free MGS into the inner membrane. However, both MGS and DMβCD molecules potentially permeated into the deeper region of the interior membrane, whereas βCD only adsorbed at the outer membrane surface. The interaction between secondary rim of βCD and the 1-palmitoeyl-2-oleoyl-glycero-3-phosphocholine (POPC) phosphate groups showed the highest number of hydrogen bonds (up to 14) corresponding to the favorable location of βCD on the POPC membrane. Additionally, the findings suggested that electrostatic energy was the main driving force for βCD adsorption on the POPC membrane, while van der Waals interactions played a predominant role in DMβCD adsorption. The release profile of MGS from the βCDs pocket across the lipid bilayer exhibited two energy minima along the reaction coordinate associated with the permeation of the MGS molecule into the deeper region of the POPC membrane.

## 1. Introduction

Alpha-mangostin (MGS; Figure 1a) is a major component of xanthones extracted from pericarps or fruit hulls of mangosteen (*Garcinia Mangostana* L.) [1,2], and exhibits a wide range of pharmacological properties [3,4], including antibacterial [5], antioxidant [6], anti-inflammatory [7,8], and anticancer activities [9,10,11]. More specifically, the antimicrobial activity of MGS targets the cell membrane of Gram-positive bacteria [12]. Unfortunately, the low water solubility of MGS leads to its limitation in pharmaceutical applications, particularly at high concentrations [13]. There are several methods to improve the water solubility of the lipophilic MGS molecule and to enhance the compound penetration through the biological membrane. One such powerful technique is the modification of the native MGS molecule with some functional groups, such as amines and triazoles, which leads to an amphiphilic MGS with an increased solubility [14,15]. Another approach is the construction of complex formation with cyclodextrins (CDs), which is a commonly used method to promote the water solubility of numerous hydrophobic molecules [16,17,18]. Note, that the truncated cone shape structure of CDs gives the potential to encapsulate small hydrophobic molecules and consequently improve their stability and biological activity [16,19].

Natural CDs include αCD, βCD, and γCD that consist of six, seven, and eight glucose subunits in the ring system, respectively, by which they can effectively enhance the solubility of poorly soluble compounds [18,20]. The physical properties and molecular dimensions of all three CDs are presented in Appendix A [21]. Among these native CDs, βCD is the most commonly used CD in pharmaceutical and industrial application due to its suitable cavity size to encapsulate various hydrophobic molecules and enhance their solubility.

Nevertheless, βCD has been found to have some drawbacks for use in such applications, because of its intrinsically problematic intramolecular hydrogen bonding results in a decreased water solubility [22].

To improve the βCD properties, βCD derivatives with substituents at some of the hydroxyl groups present in the βCD molecule have been shown to significantly increase the water solubility in comparison to the parent molecule. The most common substituents include methyl, sulfonyl butyl, and 2-hydroxypropyl groups [22,23,24]. These substituents are able to replace the hydrogen atoms on the βCD’s hydroxyl groups located in the secondary rim (O2 and O3) and/or primary rim (O6), as shown in Figure 1b. Among these derivatized βCDs, 2,6-dimethyl-β-cyclodextrin (DMβCD) is one of the most important βCD derivatives for forming host-guest inclusion complexes [25]. A previous experimental and theoretical study [26] revealed that DMβCD was the most efficient derivative for improving the water solubility of MGS. That computational study also revealed that there were two possible conformations of MGS inside the hydrophobic pocket of derivatized βCDs, which occurred from the distinct orientation of the xanthone core structure (defined as A- or C-MGS), as depicted in Figure 1c,d.

The CDs have not only been applied to enhance the solubility of hydrophobic compounds, but also to help to control the drug release through the biological membrane, depending on the type of the CD [27]. Fundamentally, the outer hydrophilic regions or polar functional groups on the CD molecules result in an improved adsorption of the encapsulated drug/molecule on the polar head groups of the lipid membrane [28] prior to delivering the molecule through the membrane [29]. One study also showed that CDs promoted drug transportation across the mucosal membrane layer via passive diffusion, in which the interaction of CDs with the biological membrane was suggested to be a pre-adsorption step before transferring the compound into the membrane [30].

The passive membrane permeability of compounds is one of the main factors in the development of a new drug candidate [31]. The two key factors for examining drug penetration across a biological membrane are the drug solubility in an extracellular aqueous phase and the permeable capability via the inner lipophilic membrane barrier [32]. Several research groups have extensively studied the influence of drug permeation through biological membranes as well as the impact of CDs as a drug carrier [27,33,34,35]. Interestingly, the formation of a host-guest inclusion complex between a drug-like molecule and hydrophilic CDs led to the increased permeability of the drug molecule across the membrane [36].

Currently, there are a number of methods and models that are available to predict the drug permeability across a biological membrane, such as the quantitative structure–permeability relationship models [37], solubility-diffusion theory [38], parallel artificial membrane permeation assay, and heterogeneous dielectric generalized born based models [39]. Alternatively, molecular dynamics simulation plays an important role in determining the dynamics behavior and release process of the drug-CDs complexes at the atomistic level [40,41].

In addition to conventional Molecular Dynamics (MD) simulation, the free energy calculation on the basis of MD umbrella sampling simulation is another potentially useful technique that is used to estimate the energy barrier of the compound across the membrane [42]. Based on this approach, Lopez et al. reported the ability of CDs in pulling cholesterol out of a lipid bilayer [43]. Yacoub et al. investigated the interactions of CDs on the lipid bilayers with the anticancer drug doxorubicin and the effect of the cholesterol concentration on doxorubicin penetration [44,45]. They found that an increased percentage of cholesterol content in the lipid membrane culminated in an enhanced free energy barrier for doxorubicin permeability.

Recently, information on the permeation of MGS at different concentrations at the lipid membrane was derived from MD simulation [46], where the hydrophobic association of the isoprenyl moiety of MGS with the lipid alkyl chains was the main driving force for the transfer of MGS across the lipid membrane bilayer [12]. An example of the permeability of MGS into a mixed type of membrane (i.e., palmitoyleoyl phosphatidylglycerol (POPG)/palmitoyleoyl phosphatidylethanolamine (POPE)) was determined [12]. However, the dissociation mechanism of the MGS from the CDs cavity followed by the transfer of MGS from the aqueous phase to the biological membrane has not been fully resolved yet.

In this present work, we decided to study the permeation of MGS into the inner lipid membrane as well as the adsorption of βCDs (βCD and DMβCD) with and without encapsulated MGS on the lipid surface using classical MD simulations. Afterwards, the free energy path calculations of MGS across the lipid bilayer were performed by means of MD umbrella sampling simulations.

## 2. Results and Discussion

### 2.1. Permeability of MGS, βCDs, and MGS/βCDs on the POPC Membrane

To investigate the permeability of free MGS, free βCDs, and MGS/βCDs inclusion complexes across the POPC membrane, triplicate MD simulations were performed under the *NP(z)AT* ensemble for 500 ns, where the MD results are given in Appendix A and Figure 2, Figure 3 and Figure 4. These simulations showed that the A-ring of free MGS in both forms (A- and C-MGS) firstly dipped into the POPC membrane, whilst the C-ring pointed towards the water layer (Appendix A). After the insertion process of the A-ring, the MGS molecule stably located at 0.9–1.0 nm above the center of the lipid bilayer (z = 0 nm). More interestingly, the findings suggested that the MGS molecule can penetrate into and be stable in the POPC membrane deeper than into the mixed POPE/POPG membrane by ca. 0.5 nm, as previously reported [12], and also that the MGS was rapidly inserted into the lipid acyl chain of the POPE/POPG membrane, and the isoprenyl group on A-ring of the xanthone scaffold was the first entering group and interacted with the inner lipid surface mainly through hydrophobic interactions [12]. The slight difference in permeable behavior of the free MGS across the membrane may arise from the different model used for constructing the lipid membrane compared to in this study.

In the case of the βCD molecule, one simulation showed that βCD was able to adsorb on the membrane surface, but not penetrate into the inner membrane (Figure 2b). From the distance plot in Figure 2b, it was noticed that the βCD stayed at a distance of around 2.6 nm from the lipid center over the course of the simulation time. Although βCD was capable of translocating from the water layer into a somewhat deeper region of the lipid bilayer, βCD still preferred to locate at a position close to the lipid polar head groups at 1.8 nm, as shown in Figure 2a,c, respectively. In contrast, DMβCD translocated from the outer lipid surface into the deeper hydrophobic region of the POPC membrane within the first 100 ns (1.00–1.05 nm in Figure 2d–f) in comparison to the βCD system. Thus, it implied that it was easier for DMβCD to permeate into the inner membrane than it was for βCD, most likely owing to the higher hydrophobicity of its lipid tails. However, the MD results within a time scale of nanoseconds did not reveal the complete penetration of CDs through the membrane [47], and there was no example of the permeation of CDs across the cellular membrane.

The dynamic behaviors of the MGS/βCDs complexes in the A- and C-form of MGS towards the lipid bilayer are depicted in Figure 3 and Figure 4, respectively. From the independent simulations, the inclusion complexes were all pre-adsorbed on the membrane surface and then changed their orientations along the simulation time prior to moving towards the polar head groups of the lipid membrane. For the A-MGS/βCD complex, it can be seen from the first two simulations that the inclusion complexes remained on the outer membrane surface along the simulation time (Figure 3a,b). Surprisingly, the dissociation process of the A-MGS/βCD inclusion complex was detected at the first 100 ns of the third MD simulation followed by the translocation of the MGS molecule into the interior membrane, as shown in Figure 3c. The MGS molecule penetrated into the hydrophobic region of the POPC membrane and was located at 0.92 nm above the center of lipid bilayer, while the βCD molecule remained attached to the polar head groups of the lipid membrane at 2.08 nm. This finding was also supported by a high energy barrier required for the translocation of the βCD molecule across the lipid bilayer [48].

For complex formation of A-MGS with DMβCD, the first simulation showed that the MGS molecule was spontaneously released from the hydrophobic pocket of DMβCD (Figure 3d), and then the molecule translocated into the inner membrane closer to the glycerol ester. After this dissociation process between the A-MGS and DMβCD, the MGS was positioned perpendicular to lipid head groups at 1.13 nm over the center of the lipid bilayer, whereas DMβCD was located beneath the phosphate groups at 0.87 nm. Contrastingly, in the case of the two remaining A-MGS/DMβCD complexes, MGS was not observed to dissociate from the hydrophobic cavity of DMβCD until 500 ns was reached (Figure 3e,f). Nevertheless, these inclusion complexes can translocate and equilibrate below the phosphate groups of the POPC membrane at a distance ranging from 1.00 to 1.20 nm. This observation reflected that DMβCD was a better carrier for the drug transfer into the membrane than βCD.

In order to compare the dynamical behavior between A- and C-MGS inside the βCDs cavity on the membrane permeability, the likely ability of the C-MGS/βCDs inclusion complexes to cross the lipid membrane was investigated. The starting conformation of C-MGS/βCDs was generated in a similar manner as for the A-MGS/βCDs (see above). The release of C-MGS from the hydrophobic pocket of βCDs was monitored from the calculation of the distance between the center of mass (COM) of MGS and each rim of βCDs (Figure 4). For the C-MGS/βCD complex, all three simulations suggested that the inclusion complex preferred to adsorb on the POPC surface rather than penetrate deeply into the interior membrane, as depicted in Figure 4a–c. Along the 500-ns MD simulations, the C-MGS/βCD inclusion complex embedded at the interface between the water and the outer lipid phases of the POPC membrane. The distance analysis indicated that the location of the inclusion complexes ranged from 1.80 to 2.70 nm above the middle of the lipid bilayer. Notably, the βCD appeared to have no ability to release the C-MGS from its cavity into the inner lipid membrane, in contrast to the A-MGS case mentioned earlier. For C-MGS/DMβCD, it was evident that this inclusion complex penetrated irrelatively deep into the hydrophobic region of the POPC membrane but did not penetrate within the first 100 ns of MD simulations (Figure 4d–f). Afterwards, the translocating behavior of the complex was proceeded by rotating their conformation followed by turning the primary rim of βCDs (P_βCD_) to point towards the polar head groups of the lipid membrane, while the secondary rim of βCDs (S_βCD_) was exposed to the water layer. After the conformational change, the inclusion complex stayed underneath the phosphate groups of the lipid bilayer at a distance of 0.80–1.00 nm. However, the MGS could not dissociate from the hydrophobic cavity of the DMβCD in all simulated systems. The last MD snapshots for each system are presented in Appendix A.

Therefore, it can be summarized here that the release processes of MGS into the POPC membrane was strongly dependent on the conformation of MGS (i.e., A- or C-form) inside the βCDs cavity. Complexation of MGS with the C-form was likely to cause a higher steric hindrance than that in the A-form for the dissociation process, and so it was rather difficult to deliver MGS from the βCDs pocket to the membrane. Similarly, a previous investigation revealed that DDT, one of the most abused insecticides, did not dissociate from the βCD cavity at the lipid membrane surface and that the DDT/βCD inclusion complex was more favorable to locate at the interface of the outer membrane region rather than permeate into the inner membrane [20].

### 2.2. Intermolecular Interaction between CDs and MGS/CDs on the POPC Membrane

To determine the interaction of free βCDs with the lipid surface, the number of hydrogen bonds (H-bond) between each rim of βCDs and the lipid head groups (i.e., phosphate and glycerol ester groups) was monitored and plotted versus the simulation time (Figure 5). The triplicate MD simulations showed that the S_CD_ of the native βCD was the main area to interact with the polar head groups of the POPC membrane via H-bond formations (Figure 5a). A number of H-bonds formed between the S_CD_ of βCD and the phosphate groups of POPC were up to 14, while the P_CD_ of βCD displayed a relatively low number of H-bonds. Moreover, the number of H-bonds between the S_CD_ of βCD and the glycerol ester groups was increased up to 4, possibly after the βCD adsorption on the POPC membrane and rearrangement of its orientation to firmly form with the interior membrane. Instead, the S_CD_ of DMβCD only formed H-bonds with the glycerol ester groups of the POPC membrane, corresponding to the more stable orientation and the deeper position of DMβCD inside the inner membrane (Section 2.1).

Apart from the adsorption of free βCDs on the lipid surface, the interaction of the MGS/βCDs complexes with the lipid membrane was also investigated by the H-bond analysis. The results suggested that both the A- and C-MGS inclusion complexes exhibited H-bond interactions of S_CD_ with the polar head groups of the POPC membrane due to the adsorption process onto the lipid membrane, like in the case of their free form (Figure 6a–d). Specifically, the third MD simulation of A-MGS/βCD, as depicted in the bottom panel of Figure 6a, suggested that the MGS molecule can form H-bonds with the glycerol ester groups better than the other simulations, probably owing to the dissociation of MGS from the βCD cavity into the hydrophobic region of the POPC membrane, as supported from the distance analysis in Figure 3c. For A-MGS/DMβCD in Figure 6b, the major H-bond interactions came from the interactions of the glycerol ester groups with the S_CD_ of DMβCD and the MGS molecule, reflecting the permeation of the inclusion complex into the inner membrane. In addition, one simulation revealed no H-bonds between MGS and DMβCD, as a result of the release of MGS out of the DMβCD cavity (top panel of Figure 6b), while the H-bond interactions between C-MGS and βCDs still existed. These results were related to the higher steric hindrance of C-MGS/βCDs on the lipid surface than that of A-MGS/βCDs, leading to a lower probability of the MGS molecule moving out of the βCDs hydrophobic pocket. Therefore, we concluded that the number of H-bonds for each system corresponded to the preferential position on the lipid bilayer. The S_CD_ was the main interacting region for forming H-bonds with the polar head groups of the POPC membrane. Moreover, we confirmed that the A-MGS conformation in complex with βCDs was more likely to be liberated from the βCDs cavity and then inserted into the inner membrane than the C-form of MGS.

### 2.3. Interaction Energy

To determine the key interaction energy of the inclusion complexes with the lipid membrane, the time evolution of the non-bonded electrostatic (*E*_EEL_) and van der Waals (*E*_vdW_) interactions between the lipid head groups and the MGS inside two βCDs was plotted (Figure 7). The results showed that the *E*_EEL_ interaction was the main interaction of the two (A/C)-MGS/βCD inclusion complexes with the lipid membrane (Figure 7a,c). The *E*_EEL_ energy between the βCD and phosphate groups of the POPC membrane had an average value of −125 kcal/mol, which was much less than the *E*_vdW_ energy. However, the interaction energy between βCD and the glycerol polar groups of the POPC membrane could not be observed. In the case of the MGS molecule, it was also more favorable to interact with the phosphate groups of the lipid membrane than with the glycerol ester groups via a similar magnitude of both *E*_EEL_ and *E*_vdW_ interactions. Conversely, the MGS and DMβCD molecules in the MGS/DMβCD complex formation interacted with both the phosphate and glycerol ester groups of the POPC membrane. The *E*_vdW_ interaction played a major role in the binding of (A/C)-MGS/DMβCDs to the lipid bilayer (Figure 7b,d). The time-dependent plots for the non-bonded interaction energies suggested that the *E*_vdW_ energy of the glycerol ester groups with DMβCD and MGS slowly decreased over the simulation time. This was the result of the permeation of the complexation into the inner membrane (glycerol ester groups). Altogether, these findings were associated with the predominant location of the complex on the lipid membrane, which was consistent with the other structural analyses (Section 2.1 and Section 2.2).

### 2.4. The PMF Calculation

Since the MD simulations on a time scale of nanoseconds did not directly reveal the energy profile for the penetration of MGS across the lipid bilayer, the release profile of MGS was further studied using the MD umbrella sampling technique. The free energy profile of MGS across the POPC membrane was investigated by means of the potential mean of force (PMF) calculations as shown in Appendix A. The results showed that there were two energy minima along the reaction pathway as the MGS molecule was moved from the water phase to the lipid phase. From the energy profile plot, the free energy value was observed to gradually decrease until reaching the first local minima at z = −0.9 nm with an energy barrier of −17.0 kcal/mol. Afterwards, the energy increased to −13.0 kcal/mol at the position close to the center of lipid bilayer. This suggests that the free energy barrier required for penetration of MGS across the inner membrane was around 4.0 kcal/mol. Although the shape of the free energy profile of MGS was somewhat different from the previously reported permeation of doxorubicin through a DPPC membrane, the free energy barrier and the local energy minima were quite similar [49]. In addition, the results of this study indicated that the calculated free energy of the MGS molecule across the POPC membrane showed a lower free energy barrier of 7.0 kcal/mol as compared to the free energy minima of MGS across the mixed membrane (POPE/POPG), possibly due to the influence of the different type of lipid membrane [46].

Currently, there are several studies on the translocation and interaction of a small molecule in the free form across a lipid bilayer [41,49,50], but the release mechanism of these molecules from the βCDs cavity is not fully understood. To investigate the release behavior of MGS/βCDs complexes into the inner leaflet of the lipid bilayer, the free energy profile for delivering the MGS molecule from the CDs pocket through the lipid bilayer was computed and averaged from three independent simulations. The average energy profiles for the system of MGS inside the hydrophobic pocket of βCD and DMβCD are depicted in Figure 8, whereas the energy profiles for each replica are shown in Appendix A. In addition, the representative structures of each stationary point, defined as (1)–(7) in Figure 8a,b, are illustrated in Appendix A.

As seen in Figure 8a, the movement of MGS out of the βCD pocket exhibited two energy minima along the reaction path. The first energy minimum (z = −1.0 nm, E = −2.5 kcal/mol) was found when the MGS molecule was initially released from the cavity of βCD (Appendix A). Afterwards, the energy value was increased to 7.0 kcal/mol at the position nearby the bilayer center (z = −0.2 nm). After passing through this point, the free energy value dropped to −1.0 kcal/mol approaching the second energy minimum at z = 0.8 nm prior to translocating the MGS molecule across another site of the lipid membrane. In the final step, the free energy was slowly elevated up to 16 kcal/mol when reaching the water phase, which reflects the unfavorable region of the MGS molecule due to its relatively high hydrophobic environment at the carbon tails. Additionally, the high number of H-bonds between βCD and the phosphate groups was observed along the releasing pathway. This observation confirmed that the βCD can only adsorb on the lipid surface, but the MGS molecule can permeate and stay in the hydrophobic region of the POPC membrane.

Similarly, for the MGS/DMβCD complex, the first energy minimum was detected at z = −0.9 nm with an energy value of −2.0 kcal/mol (Figure 8b). Afterwards, the free energy value was increased to 4 kcal/mol when the MGS molecule located close to the center of lipid bilayer (z = 0.2 nm). After the permeation of MGS across the bilayer center, the second energy minimum was found at z = 1.1 nm with an energy barrier of 1 kcal/mol. To push the MGS through the lipid bilayer, the high energy barrier (16.5 kcal/mol) was investigated in the aqueous phase, similar to that seen in MGS/βCD system. These results are also in good agreement with the energy profile of MGS passing through the lipid bilayer, as mentioned above.

Although the permeation of MGS through another site of the lipid bilayer tends to be impossible due to the high energy barrier observed in our calculations, this information is still valuable to gain a detailed insight into the release process of the MGS molecule from the βCDs cavity through the lipid membrane. In fact, in this work, we used the simplest model to study this system, which might not be enough to give the whole picture. Thus, a more realistic model (e.g., the type of membrane, cholesterol content, number of molecules, and so on) should be taken into account in the future.

Recently, the free energy landscape calculations suggested that the presence of the isoprenyl groups on the xanthone core structure of MGS was related to the higher affinity of MGS into the lipid bilayer. As such, the free energy required for transfer of MGS into the POPE/POPG membrane was smaller than that for the parent xanthone [46]. Further investigation of the free energy profile of the MGS molecules across the bacterial POPE/POPG membrane starting from the bulk water to the center of the membrane revealed that the MGS molecule could easily penetrate into the hydrophobic region of the bacterial membrane with an energy minima of 10.0 kcal/mol [14]. Moreover, determination of the free energy profile of the five drug-like molecules of caffeine, chlorzoxazone, coumarine, ibuprofen, and debrisoquine, as well as their metabolites, across the membrane indicated the continued decrease of the free energy profile from the water phase to the lipid phase. The drug-like molecules were shown to have a lower energy barrier than their metabolites, and these molecules had a higher affinity to permeate into the deeper region of the membrane than their metabolites [51]. Similar to our results, the polarity and/or H-bond interaction between small molecules and the membrane was the main driving force for the drug permeability [52].

## 3. Materials and Methods

### 3.1. System Preparation

To investigate the adsorption behavior of the focused molecules, including free-MGS, free-CDs, and the MGS/CDs complexes on the biological membrane, each molecule was initially placed at different distances (2.0–2.5 nm) from the center of the 1-palmitoeyl-2-oleoyl-glycero-3-phosphocholine (POPC) membrane (z = 0 nm). The illustration of simulation models and the details of each system are shown in Figure 9 and Table 1, respectively. The MGS was differently orientated in the aqueous phase with two distinct directions (A- or C-MGS) at 2.5 nm from the center of the POPC along the z-direction. For the free βCD and DMβCD molecules, they were located in the water phase within a range of 2.0–2.3 nm, such that the secondary rim stayed close to the polar head groups of the lipid bilayer. Likewise, the MGS/βCDs inclusion complexes (A-MGS/βCDs and C-MGS/βCDs) were positioned on the polar head groups of the lipid membrane at 2.0 nm along the z-direction. Noticeably, these conformations were generated by assuming that the complexes were adsorbed on the POPC surface prior to releasing the guest molecule into the inner membrane. In this study, the simulation models were constructed using the CHARMM-GUI membrane builder [53,54,55], and consisted of the hydrated lipid bilayer (128 POPC membrane with 64 POPC per leaflet) and 9140 TIP3P water molecules.

### 3.2. The MD Simulations

The MD simulations were performed using the GROMACS v5.1.2 software package [56,57,58], where the parameters for all atomistic models based on the AMBER force field were converted to the GROMACS format using ACPYPE.py [59]. The details of the force field parameters applied to CDs, lipids, and MGS are described in the Appendix A. To eliminate bad atomic contacts, all systems were initially minimized by the steepest descent algorithm until convergence was achieved. Then, 500-ns simulations were performed in triplicate simulations keeping the number of particles, temperature, and pressure constant (*NP(z)AT* ensemble). The periodic boundary condition was applied in all directions. The integrated time step was set to 2 fs and trajectories were collected every 10 ps. The Nosé-Hoover thermostat [60,61] was used for temperature control, while the Parrinello-Rahman barostat [62] with semi-isotropic coupling was applied for pressure control with a time constant of 3 ps and compressibility of 4.5 × 10^−5^ bar^−1^. In this study, the temperature and pressure were held constant at 298 K and 1 bar, respectively. The LINCS algorithm [63] was used to constrain all bonds involving hydrogen atoms. To treat long-range electrostatic interactions, the particle-mesh-Ewald (PME) summation method [64,65] was applied. The cut-off distance for non-bonded Coulombic and van der Waals interactions was set at 1.2 nm.

### 3.3. Free Energy Profile

Since the A-ring of MGS has a better possibility to translocate into the interior membrane, then the three different conformations of MGS (free A-MGS, A-MGS/βCD, and A-MGS/DMβCD) were selected for investigation based on potential mean of force (PMF) calculations. The schematic models and the details of each simulated system are summarized in Figure 10 and Table 2, respectively. The initial structure was built using distance-restrain and the pull module of GROMACS, in which the reaction coordinate was generated every 0.1 nm. In this study, the PMF calculation of the free A-MGS molecule contained 61 umbrella windows individually ranging from the aqueous phase through the lipid bilayer, while 56 independent simulations for A-MGS/βCD and A-MGS/DMβCD complexes were performed in three separate trials. For the harmonic potential restraint, the force constant of 1500 kJ/mol⋅nm^−2^ was only applied to the COM of MGS in the z-direction, allowing MGS to freely rotate in the xy-plane. The pressure and temperature were controlled using a Nosé-Hoover and Parrinello-Rahman approach, respectively. A restrained point was initially equilibrated for 10 ns in the *NP(z)AT* ensemble. After the equilibration step, each of the MD samplings was further collected for 20 ns as the production phase. For analysis, the PMF calculations were estimated from the last 5 ns of each window using the weighted histogram analysis method (WHAM) [66], while the Bayesian bootstrap analysis [67] (*N* = 200) was applied to estimate the statistical error.

## 4. Conclusions

In this study, the release behavior of MGS from the hydrophobic pocket of βCDs (βCD and DMβCD) was observed by the two approaches. Initially, MD simulations were applied to investigate the adsorption of free-MGS on the lipid surface in comparison with that for the MGS complexed with βCDs. By considering the different starting conformations (A- or C-form), the A-ring of MGS was first entered into the polar head groups of the POPC membrane, and stably located underneath the polar head groups of the lipid membrane. The native βCD prefers to interact with the membrane surface mainly through H-bond interactions rather than penetrate into the inner membrane. In contrast, DMβCD penetrates rather deep nearby the acyl groups of the POPC membrane. Triplicate MD simulations over a period of 500 ns showed that MGS was most likely to spontaneously release from the cavity of both βCDs, in which the dissociated MGS was then equilibrated underneath the polar head groups of the POPC membrane in the same position as the free MGS. In addition, the free energy profiles for the release of the A-MGS/βCDs inclusion complexes across the lipid membrane indicated that two energy minima were determined during the MGS translocation process. The MGS molecule dissociated from the βCDs pocket and permeated into the inner region of the POPC membrane with a relatively low energy barrier, although it appeared to be impossible to translocate MGS across the other side of the lipid bilayer due to the high energy barrier. However, our theoretical calculations also shed light on the delivery of MGS by βCD and DMβCD at the cellular membrane.

## Figures and Tables

**Figure 1 molecules-25-02532-f001:**
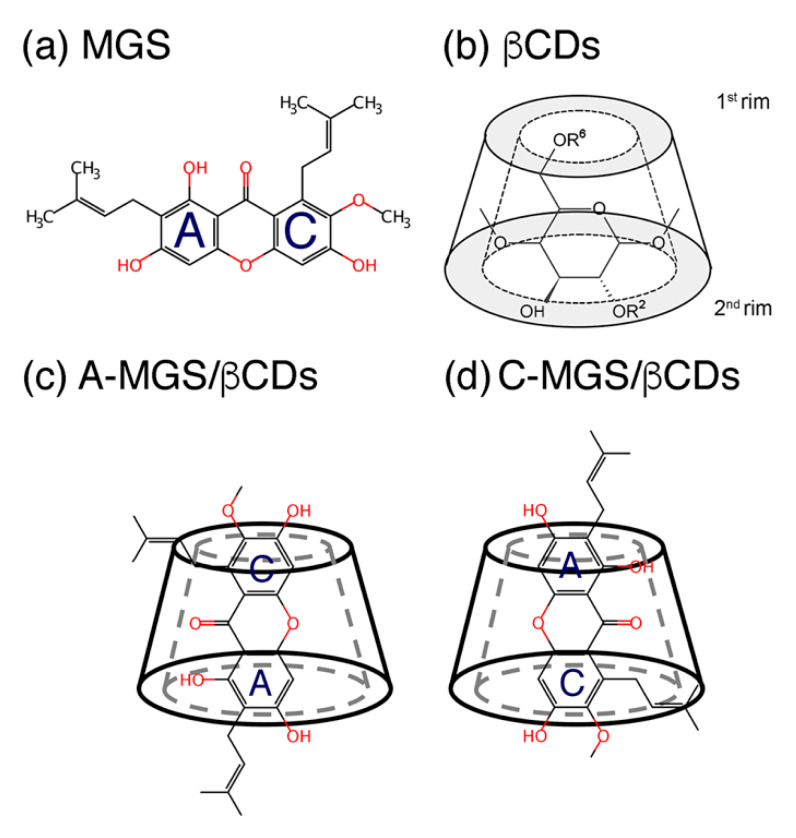
Chemical structure of (**a**) MGS, and schemes of (**b**) βCD and DMβCD used in this study, where R represents the hydrogen atoms for βCD and the methyl groups for DMβCD, as well as two orientations of MGS inside the βCD cavity referred to as the (**c**) A-form and (**d**) C-form.

**Figure 2 molecules-25-02532-f002:**
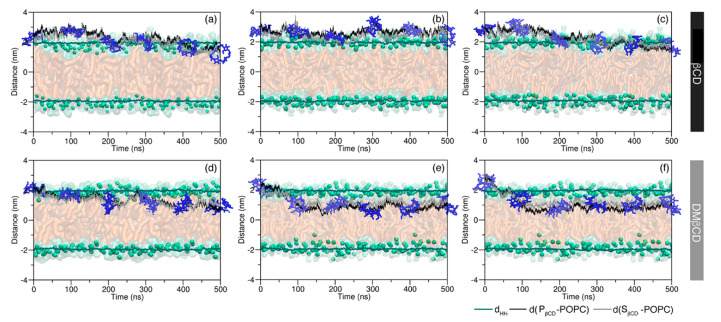
Time evolution of the distance between the centers of mass of the lipid bilayer (z = 0) and each rim of (**a**–**c**) βCD and (**d**–**f**) DMβCD. The distance between phosphate groups of each membrane leaflet (d_HH_) and the center of membrane is represented by a green line, while the distances of the primary and secondary rims of βCDs, d(P_βCD_-POPC), and d(S_βCD_-POPC), relative to the lipid bilayer are represented by black and grey lines, respectively. The MD snapshots of βCDs along the simulation time are also given in the presence of the membrane scheme.

**Figure 3 molecules-25-02532-f003:**
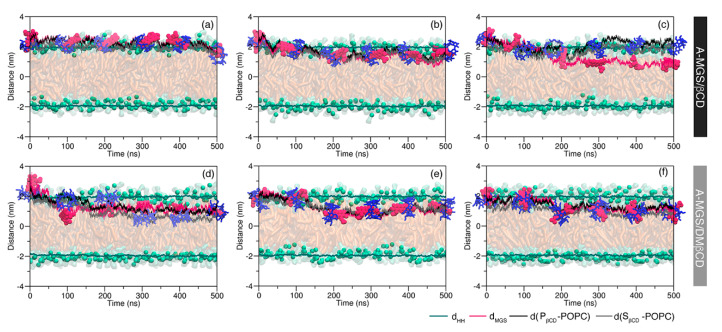
Time-dependent distance between the centers of mass of the lipid bilayer and the inclusion complexes of the A-form: (**a**–**c**) A-MGS/βCD and (**d**–**f**) A-MGS/DMβCD. The distance between the phosphate groups of each leaflet (d_HH_) is represented by the green line, while the distances of MGS as well as the primary and secondary rims of βCD, d(P_βCD_-POPC), and d(S_βCD_-POPC), relative to the lipid bilayer are represented by magenta, black, and grey lines, respectively. Representative snapshots of A-MGS/βCDs during the simulation are also given in the presence of the membrane scheme for simplicity.

**Figure 4 molecules-25-02532-f004:**
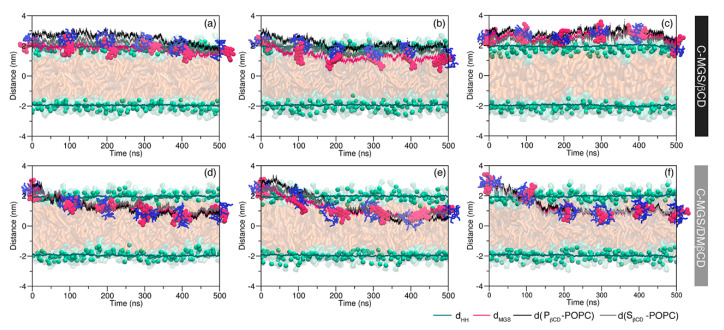
The distance of the center of the lipid bilayer (z = 0 nm) and inclusion complexes of C-MGS as (**a**–**c**) C-MGS/βCD and (**d**–**f**) C-MGS/DMβCD. The average distance between the phosphate groups of each membrane leaflet (d_HH_), the distance of the center of the lipid bilayer to the COM of MGS (d_MGS_), primary rim of βCD (d(P_βCD_-POPC)) and secondary rim of βCD (d(S_βCD_-POPC)) are represented by green, pink, black, and grey lines, respectively.

**Figure 5 molecules-25-02532-f005:**
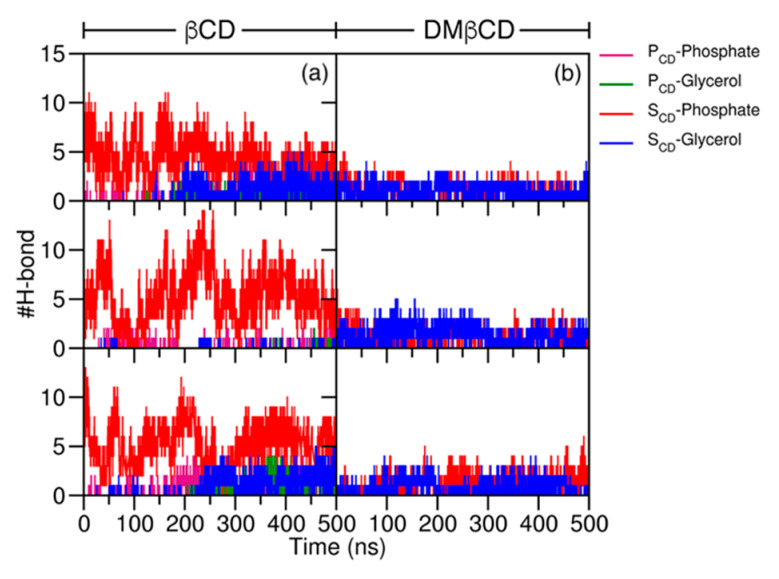
The number of H-bonds between the primary (P_CD_) or secondary (S_CD_) rim of (**a**) βCD, (**b**) DMβCD, and lipid head groups (phosphate and glycerol esters).

**Figure 6 molecules-25-02532-f006:**
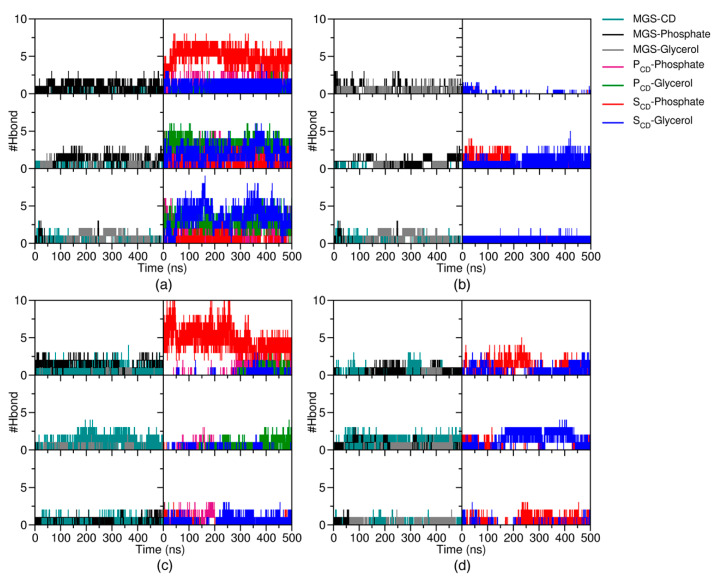
The number of H-bonds in triplicate MD simulations for each inclusion complex of (**a**) A-MGS/βCD, (**b**) A-MGS/DMβCD, (**c**) C-MGS/βCD, and (**d**) C-MGS/DMβCD.

**Figure 7 molecules-25-02532-f007:**
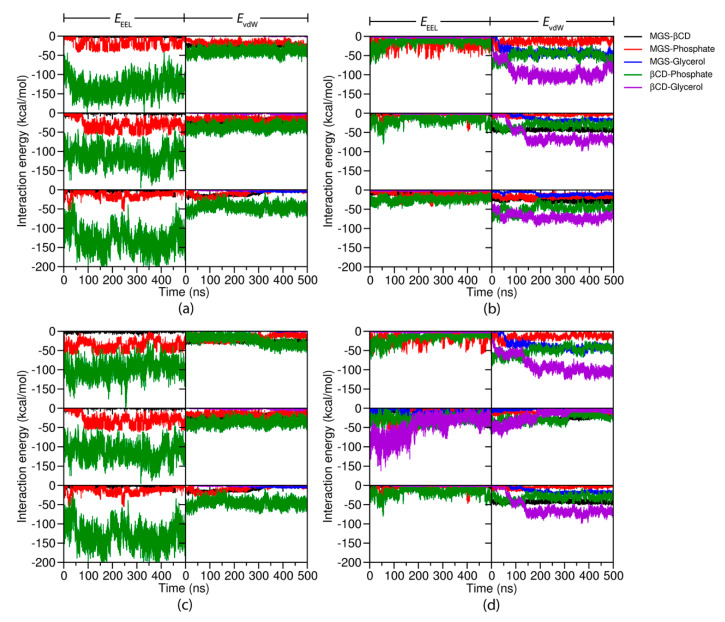
The electrostatic (*E*_EEL_) and van der Waals (*E*_vdW_) interactions for (**a**) A-MGS/βCD, (**b**) A-MGS/DMβCD, (**c**) C-MGS/βCD, and (**d**) C-MGS/DMβCD.

**Figure 8 molecules-25-02532-f008:**
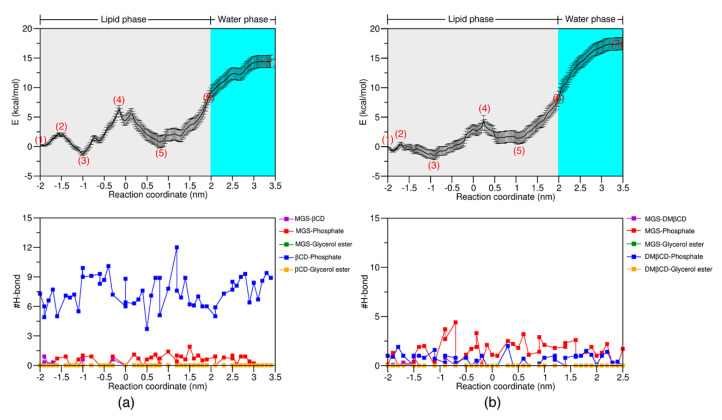
The free energy profile and time evolution of the number of H-bonds for (**a**) A-MGS/βCD and (**b**) A-MGS/DMβCD.

**Figure 9 molecules-25-02532-f009:**
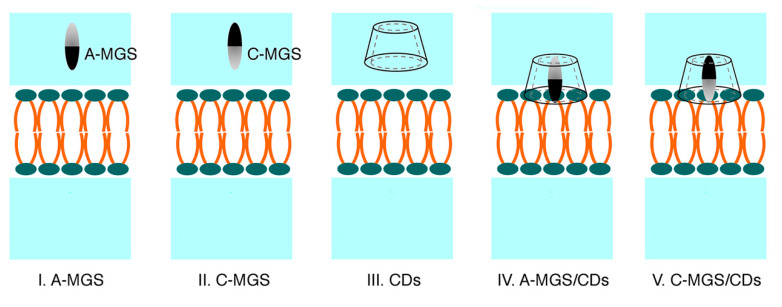
Illustration of the five studied models: (I) A-MGS, (II) C-MGS, (III) βCDs, (IV) A-MGS/βCDs, and (V) C-MGS/βCDs. Note that the A- and C-ring of MGS was represented by black and grey, respectively, whereas the βCDs (βCD and DMβCD) served as the host molecules in this study.

**Figure 10 molecules-25-02532-f010:**
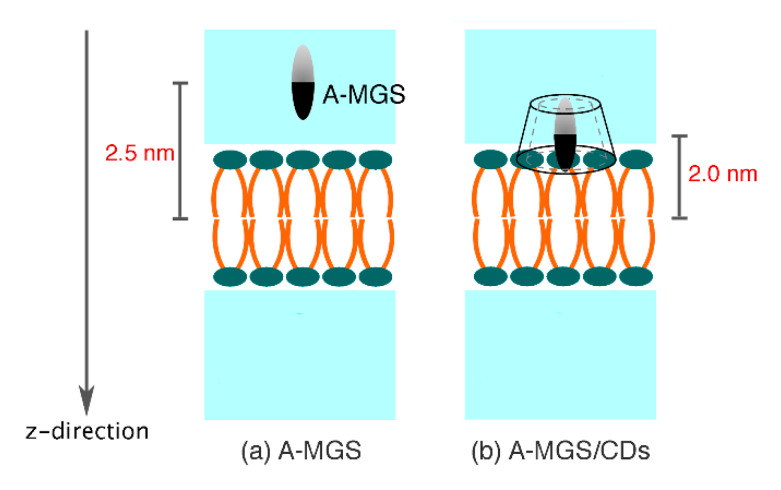
Schematic models of each system for MD umbrella sampling simulations: (**a**) free A-MGS is initially set by placing in the water phase, while (**b**) the inclusion complexes were positioned at the polar head groups of the POPC membrane.

**Table 1 molecules-25-02532-t001:** Details of the five models of free-MGS, free-βCDs, and MGS/βCDs complexes. The number of component molecules, the starting distance of the focused compound from the center of the POPC membrane bilayer (z = 0 nm) and simulation box dimensions in each system are also given.

Model	Name	Number of Molecules (MGS:CD:POPC)	Starting Distance (nm) *^a^*	Box size (nm × nm × nm)
I	A-MGS	1:0:128	2.5	6.51 × 6.29 × 10.46
II	C-MGS	1:0:128	2.5	6.60 × 6.37 × 10.23
III	βCD	0:1:128	2.3	6.52 × 6.30 × 10.48
	DMβCD	0:1:128	2.3	6.51 × 6.28 × 10.52
IV	A-MGS/βCD	1:1:128	2.0	6.39 × 6.33 × 10.65
	A-MGS/DMβCD	1:1:128	2.0	6.52 × 6.33 × 10.47
V	C-MGS/βCD	1:1:128	2.0	6.48 × 6.23 × 10.61
	C-MGS/DMβCD	1:1:128	2.0	6.52 × 6.32 × 10.47

*^a^* The distance between the focused compound and POPC membrane was calculated from the difference in their centers of mass along the z-direction.

**Table 2 molecules-25-02532-t002:** Details of the overall simulation models for the potential mean of force (PMF) calculations of the A-MGS, A-MGS/βCD, and A-MGS/DMβCD.

Model	# Windows	# Simulations	Equilibration (ns)	Sampling (ns)	Total (ns)
A-MGS	61	1	10	20	1220
A-MGS/βCD	56	3	10	20	1120 (× 3)
A-MGS/DMβCD	56	3	10	20	1120 (× 3)

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
