# Peer review of "Delivery of Alpha-Mangostin Using Cyclodextrins through a Biological Membrane: Molecular Dynamics Simulation"

_molecules, 2020, doi:10.3390/molecules25112532_

Round 1

Reviewer 1 Report

This paper is interesting and well organized. Therefore, it is recommended to be published. I only have a few comments.

  1. Please add the molecular dimensions of MGS and betaCD.
  2. As the authors calculated the PMF, please also show the free energies.
  3. The authors mentioned that they used semi-isotropic coupling on pressure control. So I guess the ensemble they used is NP(z)AT ensemble, not NPT ensemble.

Reviewer 2 Report

The authors studied the release of the α-Mangostin from the two different βCDs (βCD and DMβCD) across the lipid bilayer by molecular dynamics simulations and free energy calculations. The findings of the manuscript could be useful for the understanding and design of effective drug delivery systems. The manuscript is clear, I have only one suggestion for the authors. Since the conventional molecular dynamics simulations shows that A-MGS/DMβCD complexes could translocate and equilibrate below the phosphate groups of POPC membrane at a distance ranging from 1.0–1.2 nm, another free energy calculation of A-MGS/DMβCD might need to be conducted. Namely, A-MGS releasing starts at the position below the phosphate groups of POPC membrane instead of the surface of the membrane. Several minor things: 1. Figure 5, label (a) and (b) are missing on the plot, 2. Line 413, ‘C-MGS/DMβCD’ should be ‘A-MGS/DMβCD’, 3. Figure S4, ‘(d)-(f) A-MGS/βCD’ should be  ‘(d)-(f) A-MGS/DMβCD’, 4. Figure S5, ‘(d)-(f) C-MGS/βCD’ should be ‘(d)-(f) C-MGS/DMβCD’.

Reviewer 3 Report

This manuscript reports a very interesting simulation study of the insertion of alpha-mangostin in a phospholipid membrane either as a free molecule, or encapsulated into a cyclodextrin carrier (both a native and a chemically-modified cyclodextrin). The adopted molecular dynamics simulations were of a very high quality, of great length and carried out in triplicate. The discussion of the results is very clear, and the conclusions are sound and of great interest. In conclusion, this is a very nice manuscript reporting a high-level study, and accordingly it should be published in the present form.
